# ITMPRec: Intention-based Targeted Multi-round Proactive Recommendation

## Abstract

Personalized user preference driven recommendations have seamlessly intertwined with our daily lives. However, item providers may expect specific items to gradually increase their appeal to users over the course of users' long-term interactions with the system, but few studies pay attention to this problem. In this paper, we propose a novel intention-based targeted multi-round proactive recommendation method, dubbed ITMPRec. Specifically, we first choose a set of target items from the target category, by conducting a pre-match strategy. Afterward, we utilize a multi-round nudging recommendation method, in which we design a module to quantify the intention-level dynamic evolution of users so that we could choose more appropriate intermediate items during guidance. Besides, we model each user's sensitivity to the changes in representation induced by the intermediate items they accept. Finally, we propose a design for a Large Language Model (LLM) agent as a pluggable component to simulate user feedback. This design offers an alternative to the traditional click model based on distribution, relying on the agent's external knowledge and reasoning capabilities. Through extensive experiments on four public datasets, we demonstrate the superiority of ITMPRec compared to seven baseline models. The code repository is available at https://anonymous.4open.science/r/ITMPRec-D821.

## CCS Concepts

• **Information systems → Recommender systems**.

## Keywords

sequential recommendation, proactive recommendation, LLM

**ACM Reference Format:**
Anonymous Author(s). 2024. ITMPRec: Intention-based Targeted Multi-round Proactive Recommendation. In *Proceedings of The Web Conference (Conference acronym 'WWW)*. ACM, New York, NY, USA, 12 pages. https://doi.org/XXXXXXX.XXXXXXX

## 1 Introduction

Recommendation systems have imperceptibly become a part of our daily life's rhythm, offering immense convenience and personalized services from fashion coordination [26] to information acquisition [39], from recruitment selection [27] to travel planning [12, 15]. Traditional recommendation systems, specifically

sequential recommendation (SR) systems [4], aim to model users' historical behavior sequences, and based on this, uncover users' past preferences and interests, thereby making recommendations for items or contents in the next moment. However, if we always pander to users' historical preferences, it may harm customers and content ecosystems, in the long run [33]. For instance, as shown in Figure 1 (a), it might induce the narrowed content exposure and further lead to filter bubble [36, 38] and information cocoon [16, 28] issues because of biased feedback loop [17]. Furthermore, from the perspective of content providers, there are times when they wish to more effectively draw users' interests towards targeted contents [47].

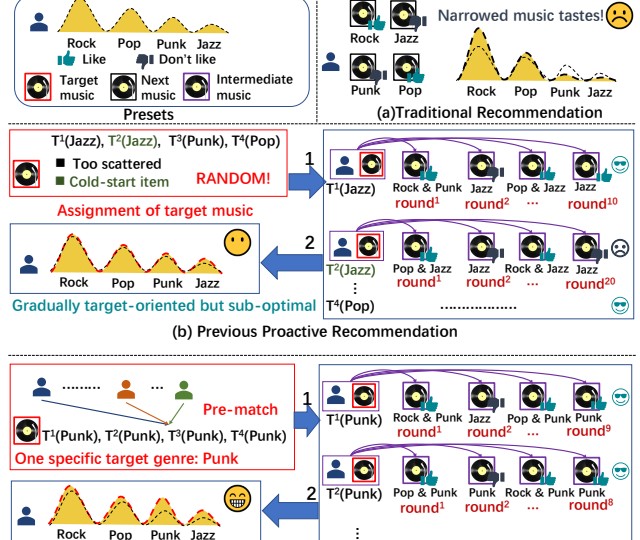

**Figure 1: The difference between traditional recommendation methods (a), previous proactive recommendation methods (b), and ours' (c).**

Therefore, there is a compelling necessity for the development of a proactive recommendation paradigm that diverges from the conventional user-centric approach. This innovative paradigm should not only capture users' preferences but also enable users to transcend the limitations imposed by their past preferences, thereby mitigating the issue of becoming ensnared in a feedback loop of homogeneity. Specifically, the system should identify some more valuable target items and, through a multi-round, progressive recommendation manner, generate different intermediate items for each user during the guidance, gradually steering the user's preferences towards the target contents.

Under such paradigm, IRN [48] first considered target item-oriented recommendation through a transformer-based framework.

Then IPG [3] was delivered by conducting proactive recommendation employing the post-processing scheme. Although existing studies have achieved some results, there are still notable limitations. (1) Randomly assigned target items without anchoring to a specific category may cause two problems, as shown in Figure 1 (b). First, by randomly designating target items within the global inventory, the chosen target might be too scattered. Second, random assignment of target items may include cold-start items which could further influence the nudging performance. (2) The role of user intention in the round-by-round nudging process is ignored. (3) The previous works assume either that the user will passively accept all intermediate items [48], or that they use a one-size-fits-all fixed threshold to measure the impact between users and intermediate items (simulated clicks) [3], which induces sub-optimal results.

To better understand why it's important to consider a whole category of items rather than just one, let's use the music recommendation task as an example. Previous target-driven recommendation methods may mark several items as target content by directly selecting them from the entire pool of musical works. However, this random designation approach might be too dispersed and often does not align well with the content providers' real needs. This is because, at specific times and in certain scenarios, content providers are often concerned with a special category or tag of target musical works, allowing these works to gain more exposure in a short period [5]. In our work, we focus more on limiting the target items to a specific category/genre, like shown in Figure 1 (c). By conducting a progressive, multi-round proactive recommendation paradigm, more users on the platform can be encouraged to enjoy the targeted content. During the process, users who originally only liked "Rock" or "Pop" music with a narrow range of interests can also bring the target content closer through multiple rounds of recommendations. Therefore, firstly, we explore nudging a class of target items, considering a more purposeful and focused content delivery, and compatible with single target item recommendation as well. Moreover, our work will pave the path for generalizing the concept of "category", such as a playlist created by combining different musical features (which includes a variety of musical works), or a mixed album composed of songs by artists of the same style. In this way, during specific periods (for instance, around Christmas or for afternoon tea), the sampled songs in the same "category" playlist/mixed album can be set aside as target items. Proactively recommending these items triggers the diversity and vibrancy of the platform's content.

Next, as to the second problem, it has been explored that user behavior patterns within recommendation systems are significantly shaped by their underlying intention [18]. User intentions [23] are coarse-grained aspects compared to user preferences, which have been proven to be effective in traditional next-item recommendation tasks [7, 20, 25, 30]. However, user intentions have not been explicitly modeled and appropriately applied in the proactive recommendation tasks. That is, existing nudging-based proactive recommendation methods neglected user intention modeling.

Besides, according to the *arousal theory* [2], each person's receptivity to external stimuli (i.e., the acceptance of an intermediate item) is different. The work [44] conducted diversity recommendations employing personalized diversity factors to boost the final performance. In this paper, we create a targeted arousal coefficient for each user, reflecting their unique reactions across multiple guidance sessions.

Last but not least, in a proactive recommendation task, it is essential to gather user responses among multiple rounds. So, we design a semi-simulator module that is useful for providing user feedback in each round. Specifically, in the user click submodule, one can alternatively use a distribution-based or LLM agent to collect users' feedback on recommended intermediate items.

To this end, we propose a new **I**ntention-based **T**argeted **M**ulti-round **P**roactive **Rec**ommendation (ITMPRec) method. The key contributions of this paper are summarized as follows:

- We deliver an idea of hauling a class of target items, where we designate a pre-match module to collect all users' opinions to generate candidate target items.
- We devise another two key components, including intention-induced score and targeted individual arousal coefficient for modeling dynamic user preference evolution during a multi-round proactive recommendation process.
- We provide an alternative LLM agent to simulate the real users' feedback in substitution for the traditional distribution-based click model.
- We conduct extensive experiments on four real-world datasets to demonstrate the effectiveness of ITMPRec, showing significant improvements over SOTA recommendation methods including sequential and proactive manner.

## 2 Related Work

In this section, we provide a review of related work, categorizing it into sequential recommendation and proactive recommendation.

**Sequential Recommendation (SR).** The sequential recommendation aims at modeling users' chronological behaviors and predicting users' next timestamp's interests. SASRec [19] was a classical sequential recommendation method utilizing the self-attention module to refine users' historical behaviors. MStein [13] proposed Wasserstein discrepancy measurement, and fused it into the InfoNCE framework. BSARec [35] leveraged the Fourier transform to inject an inductive bias by considering fine-grained sequential patterns. Furthermore, the intention modeling in SR task shows efficacy by capturing fine-grained information from sequential patterns [21, 25]. For instance, ICLRec [7] used K-Means to cluster the item embeddings and employed alternate calculation modes to consider the intention of the user in the recommendation task. ICSRec [30] extracted coarse-grained intention supervisory signals from all users' historical interaction sequences and then applied these information to construct auxiliary learning objectives for intention representation learning. Both SR algorithms and their more advanced counterparts, which incorporate user intentions, have consistently demonstrated better performance in previous studies. However, they are always centered on the next-item recommendation task and cater to the historical preferences of users. These user-centric recommendation strategies may inadvertently confine users within filter bubbles, as a result of ongoing, self-reinforcing feedback loops [1, 46].

**Proactive Recommendation (ProactRec).** Proactive recommendation represents a burgeoning field of study, encompassing

two principal avenues of investigation. One is preference evolution modeling; the other is user preference guiding. For the former type of method, earlier research primarily investigated the development of user preferences in response to interactions with recommendation systems, often employing simulation techniques [29]. Certain research endeavored to maximize long-term performances, as opposed to short-sighted actions, in light of changes in user preferences [41]. In regards to user preference guiding, there were dialog recommendation methods [11, 37], conversational recommendation [34, 47], and multi-modal recommendation [40] all proposed to guide the dialog/conversation toward the stated goal. However, the proactive manner in sequential recommendation scenarios was rarely explored. IRN [48] was a transformer-based proactive recommendation work, which assumed the user would passively accept all the middle items in the nudging path. IPG [3] was a model-agnostic post-processing method with a distribution-based click module. This method didn't consider real public datasets, instead generating synthetic data from the normal distribution. Compared to these two methods, we propose a method that is much closer to real-world scenarios for proactive recommendations.

## 3 Preliminaries

Table 1 provides a list of symbols that will be used in our paper. Having the user set $\mathcal{U}$, and the item set $\mathcal{I}$, we assume the user $u$'s previously interacted item sequence is $\mathcal{H}_u = \{x_1, x_2, ...x_t\}$, where items are ordered by timestamp chronologically.

**Table 1: Main notations used in this paper.**

| Symbol | Description |
|--------|-------------|
| $\mathcal{U}, \mathcal{I}$ | Sets of users, items |
| $N, M$ | the number of users, and items |
| $\mathcal{H}_u$ | the historically interacted item set of user $u$ |
| $L_u$ | the recommendation item list of user $u$ |
| $E$ | all item embeddings |
| $S_u$ | item embedding sequence of user $u$ |
| $\mathcal{L}_*$ | different loss functions |
| $N_C$ | the number of total intentions |
| $N_{can}$ | the number of candidate items |
| $N_{tar}$ | the number of target items |
| $R$ | the number of nudging rounds |
| $r$ | one specific round |
| $e_u, e_i$ | user embedding, item embedding |
| $\beta_u$ | the arousal coefficient of user $u$ |
| $i$ | the index of intermediate item |
| $j$ | the index of target item |

**Sequential Recommendation.** The sequential recommendation system analyzes users' historical preferences to predict the item they are most likely to click on in the upcoming timestamp, symbolized as $L_u = \{x_{t+1}|\mathcal{H}_u\}$. Given all user-item interactions, an item embedding layer, such as [4, 19], is applied to convert each item into a low-dimensional dense vector. These vectors are represented by an item embedding matrix $E \in \mathbb{R}^{M \times d}$, where $d$ is the dimension of the item embedding and $M$ is the number of items. Then, the item sequence in $\mathcal{H}_u$ can be represented as $S_u = [e_1, e_2, ...e_t]$, where each item embedding $e_i \in \mathbb{R}^d$.

Next, by devising a sequence encoder $SEQENC(\cdot)$, such as [19, 35], it will output a user representations $e_u$ as follows:

$$e_u = SEQENC(S_u), \quad (1)$$

where $e_u \in \mathbb{R}^d$ is the user representation in $d$-dimension. The $SEQENC(\cdot)$ can be unique for different SR methods.

**Intention-aware Sequential Recommendation.** We use the ICLRec [7] model to encode users' historical behaviors due to its intention modeling for sequential recommendations. The fused intention contrastive loss, combined with the main next item recommendation loss, forms the total loss as follows:

$$\mathcal{L}_{total} = \mathcal{L}_{NextItem} + \zeta \mathcal{L}_c, \quad (2)$$

where $\mathcal{L}_{NextItem}$ is the commonly used BPR loss [31], $\mathcal{L}_c$ denotes fused intention contrastive loss, and $\zeta$ is a trade-off hyperparameter. And then, the $\mathcal{L}_c$ unfolds as follows:

$$\mathcal{L}_c = \mathcal{L}_{ICL}(e_{u_1}, c_u) + \mathcal{L}_{ICL}(e_{u_2}, c_u), \quad (3)$$

where $e_{u_1}$ and $e_{u_2}$ represent two views of the user representation, and $c_u$ is the user's intention center vector which is a learnable tensor. In equation (3), the $L_{ICL}(e_{u_1}, c_u)$ is further calculated as follows:

$$\mathcal{L}_{ICL}(e_{u_1}, c_u) = -log \frac{exp(\psi(e_{u_1}, c_u))}{\sum_{neg} exp(\psi(e_{u_1}, c_{neg}))}, \quad (4)$$

where $\psi(x, y)$ denotes the dot product of two terms. This intention contrastive loss is based on InfoNCE [6]. It brings users with similar intentions closer and different intentions farther away, obtaining more discriminative user representation vectors. And $L_{ICL}(e_{u_2}, c_u)$ is defined similarly as above. After pre-training the ICLRec model, we obtain a representation $C \in \mathbb{R}^{N_C \times d}$ that is a handy global intention matrix of users, where each row $c_m \in \mathbb{R}^d$, for $1 \leq m \leq N_C$, represents one specific intention.

**Proactive Recommendation.** Differing from the traditional SR model, which only caters to users' historical interests, the proactive recommendation is a multi-round recommendation process [3]. Formally, given an anchored target item $e_j$, our task is to gradually and incrementally persuade platform users to ultimately accept it. In other words, suppose the initial user representation is $e_u^0$, and the user representation after the $r^{th}$ round of proactive recommendation is denoted as $e_u^r$. We require that $e_j^T \cdot e_u^r > e_j^T \cdot e_u^0$, where $e_j$ is the representation of the target item that remains unchanged during the process. Among them, all user representations are modeled using a pre-trained ICLRec model, similar to the one described in Equation (1). However, in this context, we also consider intermediate items during the guiding process, and revise the formula as follows:

$$e_u^r = SEQENC(S_u^r), r = [0, 1, 2, ...R], \quad (5)$$

where $S_u^r$ is a sequence containing intermediate items clicked during the first $r$ rounds. Having one intermediate item $i$, let $click(i) = 1$ be true when a user clicks on it and $click(i) = 0$ be true otherwise. Thus we have:

$$\mathcal{S}_u^r = \begin{cases} \mathcal{S}_u, \text{if } r = 0 \\ CONCAT(\mathcal{S}_u^{r-1}, e_{ui}^r), \text{if } r > 0 \,\&\, click(i) = 1 \\ \mathcal{S}_u^{r-1}, \text{if } r > 0 \,\&\, click(i) = 0 \end{cases} \quad (6)$$

Where the generation of intermediate items $e_{ui}^r$ for each user $u$ in the $r^{th}$ round will be explained in Section 4.3. Additionally, the click model denoted by $click(\cdot)$ will be elaborated on in Section 4.1.3 and 4.5.

## 4 Methodology

During the proactive recommendation process, since we are not able to get users' real-time feedback, we use a simulator module to collect users' feedback during the round-by-round process, which has been commonly used in previous studies [10]. From a macro perspective, we provide a detailed elaboration on the interplay between ProactRec systems and simulators, as illustrated in Figure 2.

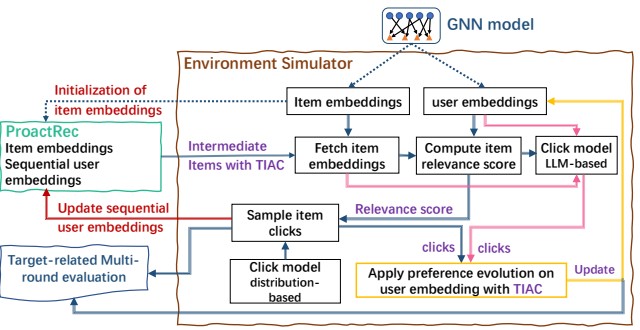

**Figure 2: The interaction illustration between simulator and recommendation method. In the figure, dashed arrows indicate that the process runs only once, while solid arrows indicate multiple rounds of execution.**

### 4.1 Environment Simulator

Conducting offline experiments poses challenges in proactive recommendation tasks, as obtaining real-time ground truth of users' internal preferences and preference changes towards an item is unrealistic. To tackle this problem, we develop an environment simulator to evaluate the effectiveness of proactive recommendation methods.

*4.1.1 User and item embeddings.* To utilize real user-item interaction dataset instead of merely synthetic datasets as in previous work [3], we use graph-based recommendation method, namely GraphAU [43] to generate pre-trained user embeddings $\hat{e}_u^0$ and items embeddings $\hat{e}_i^0$ in $d$-dimensions (dashed blue arrows in Figure 2).

*4.1.2 Preference evolution.* Users' preferences are dynamic during the interactions (the yellow part in Figure 2). After a positive interaction with item $z$ in round $r$, user $u$'s embedding will make a change. For example, $\hat{e}_u^{r+1} \leftarrow \beta_u^r \cdot \hat{e}_u^r + (1 - \beta_u^r) \cdot \hat{e}_z^r$, where $\beta_u^r$

controls the degree of preference evolution and is different for each user. It originates from the targeted individual arousal coefficient and will be expanded upon in detail in Section 4.4.

*4.1.3 Click model.* We model the interaction probability between user $u$ and item $z$ using a click model, which is always used in previous synthetic data experiments such as [9]: $a_u^r = \sigma(w((\hat{e}_u^r)^T \cdot \hat{e}_z^r - b))$, where $a_u^r$ is a binary value (1 denotes a user click and 0 otherwise), $\sigma(\cdot)$ denotes the sigmoid function, $w(\cdot)$ is the parameter of click model (such as slope and offset), and $b$ is the bias term. Alternatively, we also provide a LLM-agent click model in Section 4.5.

In what follows, we will introduce key components of our proposed method, ITMPRec, which are the pre-match module, intention-induced scores, and targeted individual arousal coefficients. The overall framework of ITMPRec is shown in Figure 3.

### 4.2 Pre-match module

In the proactive recommendation task, most of the previous work randomly assigns the target item to users [3, 48]. However, there is still ample opportunity for further enhancement in these direct strategies. In practical applications, there are often preset topics or tags that are intended for target promotion. These topics/tags encompass several target items, say $N_{can}$ of them, where $N_{can} \ll N$. Certainly, when aiming to promote a specific category of targeted items, selecting the most popular options from the candidate pools is a viable approach. However, these methods fail to address the issue where recommendation systems tend to concentrate on popular items, thereby intensifying exposure bias.

Alternatively, we calculate the overall users' preference score of the candidate target pools and then select the top $N_{tar}$, where $N_{tar} \le N_{can}$ items as the ultimate target guiding all users to get close to them. Formally, given the pre-trained user representation $e_u^0$, the target items are chosen as follows:

$$L_{N_{tar}} = cut\{sort(L_{N_{can}}, \searrow), N_{tar}\}, L_{N_{can}} = \sum_{u=1}^{U}(e_l^T \cdot e_u^0), l \in N_{can}, \quad (7)$$

where $sort(X, \searrow)$ denotes the descending sort on list $X$, and $cut\{X, num\}$ is the first $num$ elements in list $X$. Specifically, $e_j \in L_{N_{tar}}$, and $N_{tar}$ is a pre-defined value such as 20 or 50.

### 4.3 Intention-induced scores

To generate a recommendation, we employ the inner product to quantify the interaction tendency between user $u$ and item $i$ at round $r$: $score_{(u,i)}^r = (e_u^r)^T \cdot e_i$.

**Post-processing strategy.** Like in previous work [3], we take into account both the interaction probability and the degree of nudging aggressiveness. It is formulated as:

$$l_{uij}^r = score_{(u,i)}^r \cdot nudge_{(u,i,j)}^r, \quad (8)$$

where the second term is associated with the target item $e_j$ and the user $e_u^{(r+1)}$ in the subsequent round. Specifically, it is defined as $nudge_{(u,i,j)}^r = e_j^T e_u^{(r+1)} - e_j^T e_u^{(r)}$. In prior research [3], it was assumed that the transition of user representations from round $r$

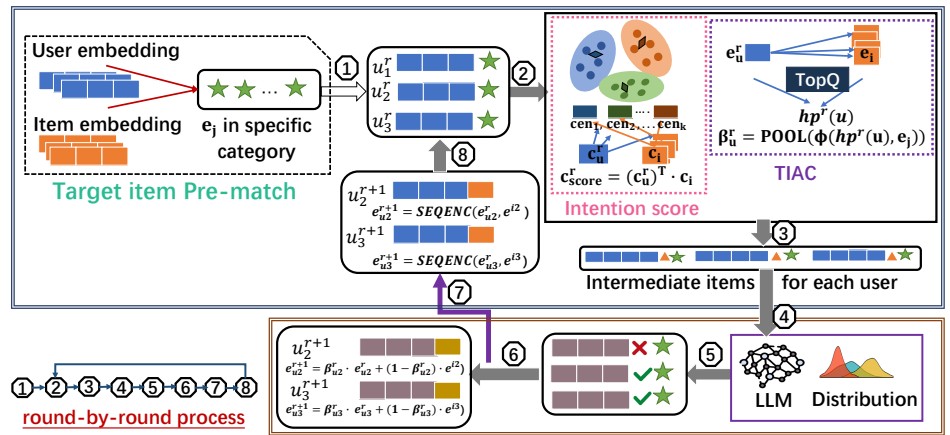

**Figure 3: The overall framework of ITMPRec**

to round $r + 1$ follows a linear pattern. For a user $u$ with all his/her intermediate items $i$, the total score can be written as:

$$l_{uij}^r = \underbrace{\{(e_u^r)^T \cdot e_i\}}_{interaction-tendency} \cdot \underbrace{\{(e_i - e_u^r)^T e_j\}}_{targeted-guidance}, \qquad (9)$$

the detailed derivation process from Equation (8) to Equation (9) can be found in Appendix A.1.

However, we believe that the intention-level score is also essential throughout the nudging process, yet it has been neglected in previous studies. Thus, we calculate the intention-level similarity between users and intermediate items to provide additional evidence for enhancing the proactive recommendation process. Particularly, having the global level intention $C \in \mathbb{R}^{N_C \times d}$ (with rows $c_1, \ldots, c_{N_C}$) from the pre-trained backbone ICLRec [7], the calculation is formulated as follows:

$$c_u^r = \underset{c_m \in \{c_1, \ldots, c_{N_C}\}}{argmin} (||c_m - e_u^r||_2^2), \qquad (10)$$

where $c_u^r$ denotes the intention-level vector of user $u$. To enable the computation between users and items, we subsequently project all items into the intention space, with the calculation detailed as follows:

$$c_i = \underset{c_m \in \{c_1, \ldots, c_{N_C}\}}{argmin} (||c_m - e_i||_2^2), \qquad (11)$$

thus the intention-level score between the user and candidate intermediate items is:

$$c_{score} = (c_u^r)^T \cdot c_i \qquad (12)$$

Thereafter, to incorporate the effect of intention, we keep the second term of Equation (9) unchanged, and modify the first term to consider not only the representational similarity between user $u$ and intermediate item $i$, but also the similarity after projection into the intention space. So, the final formulation is as follows:

$$l_{uij}^r = \langle (e_u^r)^T \cdot e_i + \lambda \underbrace{(c_u^r)^T \cdot c_i}_{intention-induced} \rangle \cdot (e_i - e_u^r)^T e_j, \qquad (13)$$

where $\lambda$ controls the weight between the intention-induced score and the interest score during the $r$ round guiding process. Then, the

item yielding the largest score in Equation (13) will be selected as the intermediate item $e_{ui}^r$ to be recommended to the user, serving as the input to the click model.

## 4.4 Targeted individual arousal coefficients

Previous work assumes that the user and its newly clicked intermediate items are linearly related and the combination coefficient of all users are the same [3]. However, numerous studies have examined how different users respond variously to fresh content or external stimuli [45], demonstrating varying levels of curiosity [32]. Therefore, in this paper, we introduce targeted individual arousal coefficients in proactive recommendation scenarios to account for users' varying degrees of acceptance towards new content.

Firstly, we calculate the historical preference variances to serve as the personalized coefficient for individual users as follows:

$$\mathbf{hp}^{r-1}(u) = TopQ\{\phi(e_u^{r-1}, e_{idx})\}, e_{idx} \in E \backslash \mathcal{S}_u^{r-1}, \qquad (14)$$

where $\phi(x, y)$ is the cosine similarity between $x$ and $y$, $TopQ\{\cdot\}$ returns a set of top $Q$ item embeddings (for items at index $idx$) with the highest $\phi$ values. $Q$ is the capacity of short-term preferences, a hyper-parameter that we will discuss more in Section 5.4.1. In other words, in Equation (14), we select the top-$Q$ items from the set of items that user has not interacted with before, based on the function $\phi$.

Then, based on the short-term preference of user $u$, denoted as $\mathbf{hp}^{r-1}(u)$, we can conduct a pooling operation to obtain an arousal value of current users toward target items $e_j$ as follows:

$$\beta_u^r = POOL(\phi(\mathbf{hp}^{r-1}(u), e_j)), \qquad (15)$$

where $POOL(\cdot)$ denotes the average pooling operation (It is worth noting that other pooling methods, such as max/sum pooling, or even more sophisticated approaches, can also be utilized. However, the selection or design of pooling functions is not the focus of this paper.). Subsequently, this targeted individual arousal coefficient (TIAC) is transmitted to the preference evolution module within the environment simulator.

## 4.5 LLM-based click simulation agent

Unlike previous proactive recommendation methods which only utilize the distribution-based click model [3] or assume the user will passively accept all intermediate items [48], in this paper, we also provide a substitute choice. We use the LLM agent to generate user feedback. The rationale behind choosing an LLM is rooted in its extensive external knowledge base and robust reasoning capabilities. Specifically, we choose ChatGLM3 [1] as our click model. Given the input $\mathcal{H}_u^r$ (note that if it is the initial round, then $\mathcal{H}_u^0 = \mathcal{H}_u$), the action of user $u$ is obtained as 0 (not click) or 1 (click) as follows:

$$a_u^r = LLM(\mathcal{P}_F, \mathcal{H}_u^r, NAMES(i_u^r)), \tag{16}$$

where $\mathcal{P}_F$ represents the task instruction (including few-shot examples, the prompt template can be found in Appendix A.2.), and the last term refers to the recommended intermediate item in the $r^{th}$ round for user $u$. For the convenience of collecting simulated click results, we give strict prompts to produce binary value in $\mathcal{P}_F$, which is 0 for not clicking, and 1 for clicking the current intermediate item. Then the next round user's historical sequence $\mathcal{H}_u^{r+1}$ will be updated as follows:

$$\mathcal{H}_u^{r+1} = \begin{cases} CONCAT(\mathcal{H}_u^r, NAMES(i_u^r)), \text{ if } a_u^r=1 \\ \mathcal{H}_u^r, \text{ if } a_u^r=0 \end{cases} \tag{17}$$

**Discussion about the traditional click model and LLM agent.** The traditional click model is always based on Bernoulli distribution [9]. The fundamental assumption is that the higher the score between the user and the item, the higher the probability that the user will accept the item. However, in reality, users' clicking behaviors are influenced by various factors, and a high score is not the only gold criterion to simulate whether the user will like an item at the next moment. The benefit of using an LLM lies in its ability to introduce the textual dimension and leverage its copious external knowledge and reasoning capabilities to model intricate decision-making factors for users in the present era [22].

To sum up, we provide the overall method flow in Appendix A.3.

**Table 2: Data Statistics**

| Dataset | Lastfm | ML-100k | Steam | Douban_movie |
|---|---|---|---|---|
| #Users | 945 | 943 | 12,611 | 2,623 |
| #Items | 2,782 | 1,348 | 2,017 | 20,527 |
| #Interactions | 246,368 | 98,704 | 220,100 | 1,161,110 |
| Density | 9.3712% | 7.7649% | 0.9686% | 2.1565% |
| #Avg. Items per User | 36.78 | 104.67 | 19.54 | 442.66 |

## 5 Experiments

In this section, we present detailed experiments to demonstrate the effectiveness of ITMPRec by answering the following research questions (RQs): **RQ1**: How effective are the key model components (e.g. pre-match module, intention-induced score, and targeted individual arousal coefficient) in ITMPRec? **RQ2**: How does ITMPRec perform in comparison to state-of-the-art sequential recommendation techniques and other target-driven multi-round recommendation

[1]https://github.com/THUDM/ChatGLM3

methods? **RQ3**: How is the robustness (e.g., hyper-parameters sensitivity) and scalability (combination with LLM-based click simulation agent) of our model?

## 5.1 Experiment Settings

We conducted our experiments on four publicly available datasets: Lastfm [2], ML-100k[3], Steam [4], and Douban_movie [5]. The data statistics are shown in Table 2.

**Implementation Details.** Our work is implemented by Pytorch. The embedding size $d$ is set as 64 for all methods. The total number of nudging rounds $R$ is set as 20 for all datasets. As for the target item number $N_{tar}$, it is set as 50 for the Lastfm and ML-100k datasets, and as 20 for the Steam and Douban_movie datasets. The max length of user sequences is set as 50 (like previous SR work do [7, 30]). The $\lambda$ is set by grid search techniques from the list [5, 1, 0.1, 0.01, 0.001], and then it assigns to 1 for Lastfm, and ML-100k datasets, 0.01 for Steam dataset and 5 for Douban_movie dataset. The learning rate is $1e^{-3}$. The experiments are conducted on a server with NVIDIA GeForce RTX 3090.

**Evaluation Protocols.** The performances are evaluated by the HitRatio (HR@P, it quantifies the ratio of items that receive positive interactions over $P$ proactive recommendation cycles. Formally, $HR@P = \frac{1}{P|\mathcal{U}|} \sum_{p=1}^P \sum_{u\in\mathcal{U}} a_{up}$, where $a_{up} \in \{0, 1\}$ represents the feedback from click simulator of user $u$ in round $p$.) as well as two metrics to measure the quality of proactive recommendations, as used in [3, 48], which are Increase of Interest (IoI@P), and Increase of Ranking (IoR@P). Among them, $P \leq R$, which is set as $P \in [5, 10, 15, 20]$. In other words, we evaluate in different stages in multiple rounds of recommendation. Firstly, the IoI@P is calculated as follows:

$$IoI@P = \frac{1}{|\mathcal{U}|} \sum_{u\in\mathcal{U}} (\hat{e}_j^T \cdot \hat{e}_u^P - \hat{e}_j^T \cdot \hat{e}_u^0), \tag{18}$$

where $\hat{e}_u^P, \hat{e}_u^0$ are the user embedding in the simulator at round $P$, and the start of the guidance phase, respectively. And $\hat{e}_j$ represents the target item to guide towards.

Secondly, the $IoR@P$ is calculated as follows:

$$IoR@P = \frac{1}{|\mathcal{U}|} \sum_{u\in\mathcal{U}} \mathsf{Ran}\{\hat{e}_j|\hat{e}_u^0\} - \mathsf{Ran}\{\hat{e}_j|\hat{e}_u^P\}, \tag{19}$$

where $\mathsf{Ran}\{\hat{e}_j|\hat{e}_u^*\}$ denotes the discrete ranking of the target item $\hat{e}_j$ among all items based on their similarity to the user $\hat{e}_u^*$, computed from different round of user representation $\hat{e}_u^*$.

**Baselines.** We compare our ITMPRec with two main taxonomies, including sequential recommendation methods (**SR**): **SASRec** [19], **ICLRec** [7], **MStein** [13], **ICSRec** [30], and **BSARec** [35]; as well as proactive recommendation methods (**ProactRec**): **IRN** [48], and **IPG** [3]. The details can be found in Appendix A.4.

## 5.2 RQ1: Ablation Study

We conduct an ablation study on four datasets, aiming to analyze the contribution of each component within ITMPRec. The following

[2]https://grouplens.org/datasets/hetrec-2011/
[3]https://grouplens.org/datasets/movielens/100k/
[4]https://cseweb.ucsd.edu/~jmcauley/datasets.html#steam_data
[5]https://www.kaggle.com/datasets/fengzhujoey/douban-datasetratingreviewside-information

are the three variants: **(1) w/o Pre-match (P):** ITMPRec without the pre-match module, implemented by replacing ITMPRec with a random target selection scheme from all items. **(2) w/o Intention-induced scores (IIS):** ITMPRec without the intention-induced score module. **(3) w/o TIAC:** ITMPRec without the targeted individual arousal coefficient module.

**Table 3: Results of ablation studies on four datasets**

| Dataset | Ablation | HR@20 | IoI@20 | IoR@20 |
|---|---|---|---|---|
| Lastfm | w/o P | 0.4113 | 0.6161 | 141.8555 |
| | w/o IIS | 0.3324 | 0.403 | 97.2408 |
| | w/o TIAC | 0.3758 | 0.5149 | 116.5403 |
| | ITMPRec | **0.4135** | **0.6614** | **161.7352** |
| ML-100k | w/o P | **0.4067** | 0.4622 | 131.4221 |
| | w/o IIS | 0.3878 | 0.4596 | 136.6786 |
| | w/o TIAC | 0.3823 | 0.4006 | 118.3061 |
| | ITMPRec | 0.4016 | **0.469** | **139.6954** |
| Steam | w/o P | 0.3907 | 0.3108 | 59.8572 |
| | w/o IIS | 0.3920 | 0.3321 | 71.5609 |
| | w/o TIAC | 0.3858 | 0.2472 | 38.4798 |
| | ITMPRec | **0.3923** | **0.3336** | **71.6806** |
| Douban_movie | w/o P | **0.3389** | 0.3201 | 73.9921 |
| | w/o IIS | 0.3329 | 0.3035 | 64.1521 |
| | w/o TIAC | 0.3303 | 0.2644 | 50.9361 |
| | ITMPRec | 0.3362 | **0.3374** | **77.2108** |

The results are reported in Table 3. The pre-match (P) module is designed for selecting a bunch of target items from a specific category. Collecting all users' opinions can avoid the problem induced

by the random choice of target item (which will encounter some cold-start items and lead the nudging result trap into a sub-optimal result). As observed in the table, in the Lastfm and Douban_movie datasets, the degradation of w/o IIS is significant, indicating it is imperative to model the user's intention-induced scores. It is noteworthy that in the Steam dataset, the contribution of intention-level scores is relatively limited. This is due to the inherent constraint that the number of items a user can search for is significantly smaller compared to the vast number of users. Consequently, various strategies for intermediate item selection may not exhibit substantial differences, as the pool of candidate items is restricted. The purpose of the TIAC module is to capture the individual's sensitivity to external stimuli during the proactive recommendation process. Generally, the larger the user base, the more diverse and personalized the needs of users tend to be. Therefore, when compared with the Lastfm and ML-100k datasets, the performance of the TIAC module on the Steam and Douban_movie datasets is significantly better, further indicating that capturing the personalized needs of different users is crucial in multi-round recommendation tasks.

Overall, ITMPRec has achieved significant performance on the interest nudging metric (both IoI@P and IoR@P), while the HR@P metric has slightly decreased within an acceptable range. This is because all three modules of ITMPRec are designed from the perspective of steering users towards target content, effectively enhancing the quality of proactive recommendation. The decrease in HR@P may be attributed to our guidance process using a pre-trained

**Table 4: The overall experiment results on four datasets. The bold is the best performance and the underline is the second-best.**

| Datasets | Methods | HR@5 | IoI@5 | IoR@5 | HR@10 | IoI@10 | IoR@10 | HR@15 | IoI@15 | IoR@15 | HR@20 | IoI@20 | IoR@20 |
|---|---|---|---|---|---|---|---|---|---|---|---|---|---|
| Lastfm | SASRec | 0.3263 | 0.0094 | -0.1749 | 0.3254 | 0.0174 | -0.6057 | 0.3248 | 0.025 | -0.7632 | 0.3243 | 0.0311 | -1.1204 |
| | ICLRec | 0.4137 | 0.0083 | 0.1126 | 0.4129 | 0.0102 | 0.4359 | 0.4111 | 0.0066 | 8.594 | 0.4106 | 0.0001 | 0.9521 |
| | MStein | 0.3289 | -0.0024 | -0.6893 | 0.3281 | 0.0023 | -0.8823 | 0.3275 | 0.0139 | -0.925 | 0.327 | 0.024 | -0.9139 |
| | BSARec | 0.3334 | 0.0193 | 0.5216 | 0.3327 | 0.0297 | 0.7023 | 0.332 | 0.04 | 0.6054 | 0.3315 | 0.0493 | 0.5891 |
| | ICSRec | 0.3369 | 0.0115 | -0.1695 | 0.3359 | 0.0251 | -0.0528 | 0.3351 | 0.0362 | 0.0099 | 0.3345 | 0.0458 | 0.0688 |
| | IRN | 0.4028 | 0.0101 | 0.0185 | 0.4018 | 0.0203 | 0.0916 | 0.4008 | 0.04 | 1.824 | 0.4002 | 0.0525 | 3.2734 |
| | ICLRec-IPG | 0.3516 | 0.1791 | 25.2901 | 0.3528 | 0.2976 | 52.9695 | 0.349 | 0.3879 | 80.5057 | 0.352 | 0.4544 | 100.1863 |
| | ITMPRec w/o P | **0.4163** | 0.2925 | 60.5283 | 0.4110 | 0.5218 | 120.0176 | **0.4122** | 0.5958 | 137.4319 | 0.4113 | 0.6161 | 141.8555 |
| | IMPRec | 0.4129 | **0.3943** | 96.9189 | 0.4153 | **0.5938** | 146.0627 | 0.4115 | **0.6486** | 159.1564 | 0.4135 | **0.6614** | 161.7352 |
| ML-100k | SASRec | 0.3994 | 0.0455 | -0.4036 | 0.3991 | 0.0866 | -0.9826 | 0.398 | 0.1121 | -1.2254 | 0.3979 | 0.1259 | -1.1867 |
| | ICLRec | 0.4124 | 0.0394 | 0.2398 | 0.4117 | 0.0744 | 0.2578 | 0.4102 | 0.0952 | 0.2476 | 0.4083 | 0.1052 | 0.0111 |
| | MStein | 0.3134 | 0.0074 | -0.1127 | 0.3125 | 0.0141 | -0.1355 | 0.3118 | 0.0204 | -0.022 | 0.3114 | 0.0264 | 0.12 |
| | BSARec | 0.3705 | 0.0416 | -0.3646 | 0.3702 | 0.0814 | -0.7027 | 0.3692, | 0.1131 | -1.1309 | 0.3689 | 0.1365 | -1.5442 |
| | ICSRec | 0.3642 | 0.0412 | -0.0593 | 0.3636 | 0.0866 | 0.0346 | 0.3628 | 0.1231 | 0.2145 | 0.3621 | 0.1503 | 0.2695 |
| | IRN | **0.4274** | 0.0299 | 0.0518 | **0.427** | 0.0578 | 0.2712 | **0.425** | 0.0867 | 1.3507 | **0.4237** | 0.0912 | 1.7407 |
| | ICLRec-IPG | 0.3866 | 0.152 | 33.2767 | 0.3891 | 0.262 | 68.703 | 0.3895 | 0.3409 | 96.4608 | 0.3861 | 0.3898 | 111.8751 |
| | ITMPRec w/o P | 0.4029 | 0.2353 | 63.9955 | 0.4027 | 0.3951 | 113.2598 | 0.4066 | 0.4496 | 128.8455 | 0.4067 | 0.4622 | 131.4221 |
| | ITMPRec | 0.4064 | **0.2433** | 70.0011 | 0.4024 | **0.3998** | 120.6669 | 0.404 | **0.4556** | 136.867 | 0.4016 | **0.469** | 139.6954 |
| Steam | SASRec | **0.4271** | 0.0486 | -0.2202 | **0.4263** | 0.0991 | 0.3557 | **0.4257** | 0.132 | 1.1601 | **0.4251** | 0.1521 | 1.6881 |
| | ICLRec | 0.3886 | 0.0583 | 0.8334 | 0.3878 | 0.114 | 2.0505 | 0.3872 | 0.1571 | 3.3948 | 0.3867 | 0.1898 | 4.6866 |
| | MStein | 0.3929 | 0.0584 | 1.1366 | 0.3921 | 0.1166 | 2.4076 | 0.3914 | 0.162 | 2.7133 | 0.3909 | 0.1942 | 2.5779 |
| | BSARec | 0.4096 | 0.0608 | 0.1626 | 0.4089 | 0.129 | 2.0218 | 0.4083 | 0.176 | 4.0323 | 0.4078 | 0.205 | 5.6237 |
| | ICSRec | 0.4005 | 0.0597 | 0.0492 | 0.3998 | 0.1223 | 0.7546 | 0.3991 | 0.1656 | 1.6664 | 0.3986 | 0.1927 | 2.0173 |
| | IRN | 0.4205 | 0.0418 | 0.3826 | 0.4195 | 0.0839 | 0.586 | 0.4188 | 0.1628 | 2.6768 | 0.4183 | 0.2016 | 6.626 |
| | ICLRec-IPG | 0.3921 | 0.1036 | 17.6234 | 0.3907 | 0.1777 | 27.688 | 0.3898 | 0.2245 | 33.4087 | 0.3895 | 0.2554 | 37.4944 |
| | ITMPRec w/o P | 0.3911 | 0.1876 | 46.7208 | 0.3899 | 0.2654 | 55.9344 | 0.3915 | 0.2984 | 58.9080 | 0.3907 | 0.3108 | 59.8572 |
| | ITMPRec | 0.3918 | **0.2192** | 55.3553 | 0.3937 | **0.2955** | 66.6745 | 0.393 | **0.3239** | 70.6409 | 0.3923 | **0.3336** | 71.6806 |
| Douban_movie | SASRec | 0.3673 | -0.0021 | 0.0888 | 0.3669 | -0.0042 | 0.2017 | 0.3662 | -0.0046 | 0.3321 | 0.3655 | -0.004 | 0.5044 |
| | ICLRec | 0.3277 | 0.0002 | 0.0062 | 0.3268 | -0.0017 | 0.0043 | 0.3261 | -0.0009 | 0.0475 | 0.3256 | 0.0019 | 0.175 |
| | MStein | 0.3174 | 0.003 | 0.018 | 0.3166 | 0.0076 | 0.0636 | 0.3159 | 0.0128 | 0.1195 | 0.3154 | 0.0176 | 0.2197 |
| | BSARec | **0.4217** | -0.0046 | 0.0028 | **0.4215** | -0.0095 | -0.0768 | **0.4208** | -0.013 | -0.146 | **0.42** | -0.015 | -0.2929 |
| | ICSRec | 0.3304 | 0.0019 | 0.0858 | 0.3296 | 0.0016 | 0.1511 | 0.3289 | 0.0037 | 0.2715 | 0.3284 | 0.0066 | 0.4051 |
| | IRN | 0.3758 | 0.0037 | 0.1676 | 0.3753 | 0.0069 | 0.2913 | 0.3744 | 0.0052 | 0.4284 | 0.3739 | 0.001 | 0.6543 |
| | ICLRec-IPG | 0.331 | 0.0849 | 13.3451 | 0.3323 | 0.1418 | 21.2825 | 0.331 | 0.1885 | 30.0722 | 0.3303 | 0.2259 | 39.0427 |
| | ITMPRec w/o P | 0.3439 | 0.1465 | 33.6715 | 0.3483 | 0.2222 | 48.5319 | 0.3422 | 0.2798 | 62.1714 | 0.3389 | 0.3201 | 73.9921 |
| | ITMPRec | 0.3366 | **0.1619** | 36.0797 | 0.3363 | **0.2408** | 50.5707 | 0.3361 | **0.296** | 65.3341 | 0.3362 | **0.3374** | 77.2108 |

ICLRec [7] model, which aligns with users' historical preferences. However, our target item-driven process may deviate from these preferences, yet the decrease in HR@P remains insignificant. To further enhance HR@P, we could consider incrementally updating the model. However, this may compromise time performance and introduce additional computational overhead. Balancing time efficiency and precise performance in proactive recommendation is a promising direction for future work (see Appendix A.7).

## 5.3 RQ2: Overall performance comparison

We compare ITMPRec with seven SOTA methods detailed in Appendix A.4. For the sake of fairness, we use distribution-based click simulator for all methods. The overall experiment results are shown in Table 4. From the table, we can draw the following conclusions: (1) Comparing traditional SR methods (such as SASRec, ICLRec, ICSRec) with proactive ones (i.e., IRN, IPG, ITMPRec), proactive methods outperform SR in IoI and IoR, demonstrating their superiority in progressive recommendations. SR methods often show negative IoI and IoR values, suggesting user preferences diverge from target items. Although proactive methods slightly underperform SR in HR@P due to deviating from historical preferences, the decrease is insignificant and acceptable. (2) Among proactive recommendations, IRN's guiding performance is limited due to its one-time path generation and user passivity assumption. With simulated click feedback, IRN's IoI and IoR improvements are slow. (3) Finally, both versions of our method, with and without pre-match, outperform SR and proactive recommendation methods in terms of IoI@P and IoR@P metrics. ITMPRec significantly improves over IPG, the second-ranked proactive recommendation method, with average enhancements of 36.46% in IoI@20 and 68.82% in IoR@20 across four datasets. This shows our approach effectively addresses guiding to a target category and single item.

In addition, we also provide an extended version of ITMPRec that seamlessly integrates with the LLM simulation click model. We analyze the superiority of LLM-based click simulation over distribution-based click model from quantitative and qualitative perspectives (see Appendix A.5).

## 5.4 RQ3: Parameter Sensitivity and Case Study

*5.4.1 Parameter Sensitivity Analysis.* We further investigate the impact of our model's hyper-parameters, specifically $Q$ (the number of items that will be considered as a personal curiosity in section 4.4) and $N_C$ (intention number) on four datasets. The results are shown in Figures 4 and 5, respectively. From Figure 4, we observe that due to the relative density of the Lastfm and ML-100k datasets, sampling the 5 most likely preferences and then calculating similarity scores with the target items can be enough to characterize how users respond to new external stimuli. However, for sparser datasets like Douban_movie and Steam, more of a user's preferences need to be sampled, for example, 20, to better model the user's arousal level.

As to Figure 5, the larger value $N_C$ indicates that users have a broader range of intentions. The best intention number for each dataset is unique. For smaller datasets such as Lastfm and ML-100k, the model performs best when the number of intentions is set to 32. In contrast, for larger datasets like Steam and Douban_movie, the model shows better performance when the number of user

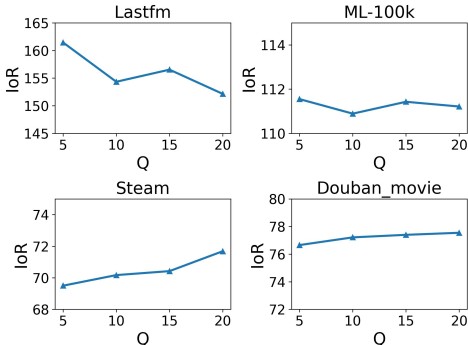

**Figure 4: Influence of the sampling number $Q$**

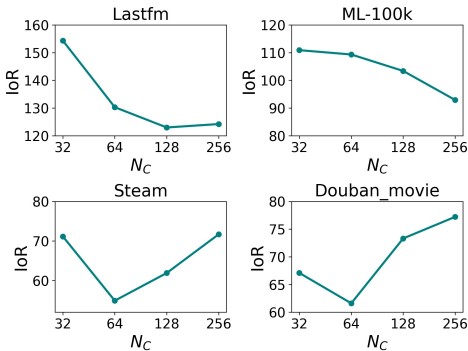

**Figure 5: Influence of the intention number $N_C$**

intentions is set to 256. In general, the smaller the dataset, the less number of intentions needed and vice versa.

*5.4.2 Case Study.* Besides, to further illustrate the effectiveness of ITMPRec, we conduct a case study (see Appendix A.6) on visualization of user embedding evolution in multiple rounds during the proactive recommendation process.

## 6 Conclusion

In this paper, we introduce ITMPRec, a novel multi-round proactive recommendation model, which breaks the limitation of traditional sequential recommendation methods that only consider user interests through a target item-driven strategy. Specifically, we focus on the problem of targeting one category of items in proactive recommendations and propose a pre-match module to effectively select target items. During the multiple-round process, we consider the importance of the intention of users (intention-induced scores) capturing the coarse-level evidence for next-round steering recommendation. In addition, since each user has a different reaction to external stimuli, we devise a TIAC module to further boost the performance. Lastly, we design a pluggable LLM-based click simulator agent, which lays on the unique strengths of LLM and will better imitate the user's feedback on intermediate items. Extensive experiments on four real-world datasets prove the superiority of ITMPRec, and the ablation study further verifies the effectiveness of each component of our model.

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

# A  Appendix

## A.1  The mathematical derivation

Given the following equation:

$$l_{uij}^r = score_{(u,i)}^r \cdot nudge_{(u,i,j)}^r, \tag{20}$$

where $score_{(u,i)}^r = (e_u^r)^T \cdot e_i$, and $nudge_{(u,i,j)}^r = e_j^T e_u^{(r+1)} - e_j^T e_u^{(r)}$.

Substitute into the formula, we get:

$$l_{uij}^r = ((e_u^r)^T \cdot e_i) \cdot (e_j^T e_u^{(r+1)} - e_j^T e_u^{(r)}) \tag{21}$$

In the previous work [3], user with the same coefficient $\omega$ combines intermediate items and the old user representations, resulting in $e_u^{r+1} = \omega e_u^r + (1 - \omega)e_i$. So Equation (21) can be rewritten as:

$$\begin{aligned} l_{uij}^r &= ((e_u^r)^T \cdot e_i) \cdot (1 - \omega)(e_i - e_u^r)^T e_j \\ &= (1 - \omega)((e_u^r)^T \cdot e_i) \cdot (e_i - e_u^r)^T e_j, \end{aligned} \tag{22}$$

in Equation (22), our goal is to select one intermediate item $i$ with the highest score of $l_{uij}^r$, and then input into the downstream click simulator. With or without $(1 - \omega)$ term cannot influence ultimate choice of $i$, thus it can be omitted and get:

$$l_{uij}^r = ((e_u^r)^T \cdot e_i) \cdot (e_i - e_u^r)^T e_j, \tag{23}$$

which is is equivalent to Equation (9).

## A.2  The prompt of LLM-based click simulation

The prompt template of the Lastfm dataset is displayed in Figure 6.

| **3-shot prompt of Lastfm** |
|---|
| You are now a recommendation agent. Given history behaviors (listened music artists were separated by ::) and you just give the answer as '1' or '0' about the final decision. '0' represents you don't accept, and '1' denotes you will accept the recommended item. Q: The person has listened some artists' music, Dawn of Ashes::The Faint::Goldfrapp::Shape of Despair. Please predict the next music artist Damn Yankees will be liked or not. A: 1 Q: The person has listened some artists' music, Apollo 440::The Young Gods::Electrosoul System::Blue Sky Black Death. Please predict the next music artist Carlos Puebla will be clicked or not. A: 0 Q: The person has listened some artists' music, Claire Pichet::Yann Tiersen & Shannon Wright::Thes One::Dday One. Please predict the next music artist Ceylan Ertem will be clicked or not. A: 1 Q: The person has listened some artists' music [HistoryHere] recently. Please predict the next music artist [CansHere] will be accepted or not. What is your answer? |

**Figure 6: The prompt example of the Lastfm dataset, which is $3-$shot manner.**

## A.3  Algorithm

The overall algorithm process of the proposed method ITMPRec is provided in Algorithm 1. Line 1 stands for the pre-match calculation. For each target item $j$, the nudging path is independent, and it is initialized uniquely for each user and target item combination on Line 3. Line 8 is responsible for obtaining the intention-level score, while Line 9 calculates the total score by selecting the $r^{th}$ intermediate item for user $u$. Line 10 retrieves the $\beta_u^r$ value. Moving on, Line 12 performs click simulation. If the intermediate items are

---

**Algorithm 1** ITMPRec

**Input:** user set $\mathcal{U}$; item set $\mathcal{I}$; historical sequences $\mathcal{S}_u$, where $u$ changes from 1 to $|\mathcal{U}|$; nudging round number $R$; batch size $B$.

**Output:** The nudging path of each user $P_{uj}^r$ for each target content $j$.

1: Get target items to be nudged by Equation (7).
2: **while** j in range $(N_{tar})$ **do**
3:    $P_{uj}^0$ = [].
4:    **for**  r in range(R) **do**
5:        Get user representation based on $\mathcal{S}_u^r$ by Equation (5).
6:        intermediate list $intermids_r$ = [], $recs_u^r$ = [].
7:        **for** $u_{batch}$ in range(U, step=B) **do**
8:            Get intention-level score in guidance by Equation (12).
9:            $rec_u^r = \arg\max_i(l_{uij}^r)$ {//Get the overall score of intermediate items by Equation (13)}.
10:            Get targeted individual arousal coefficient by Equation (15).
11:            $recs_u^r$.extend($rec_u^r$)
12:            $intermid_r = clicks\{rec_u^r, \beta_u^r\}$.
13:            $intermids_r$.extend($intermid_r$).
14:        **end for**
15:        **for** iidx in range(len($intermids_r$)) **do**
16:            **if** $intermids_r$[iidx]  **then**
17:                Update $S_u^{r+1} = CONCAT(S_u^r, recs_u^r[iidx])$
18:                $P_{uj}^r.extend(recs_u^r[iidx])$
19:            **end if**
20:        **end for**
21:    **end for**
22: **end while**
23: **return** $P_{uj}^r$

---

clicked, Line 16 updates the nudging path $P_{uj}^r$ accordingly. For each user $u$, this process, from Line 2 to Line 22, is repeated $N_{tar} \times R$ times. Finally, the algorithm returns the final nudging path.

## A.4  Detail introduction on baseline methods

Here, we provide detailed information about each of the baseline methods. They include:

**(1) Sequential Recommendation Methods (SR)**

- **SASRec** [19]: The classical sequential recommendation method with self-attention framework.
- **ICLRec** [7]: An intent contrastive learning paradigm that can model the latent intention of users and fuse them into a SR method via a new contrastive self-supervised learning objective.
- **MStein** [13]: It is a mutual Wasserstein discrepancy minimization-based sequential recommendation method.
- **ICSRec** [30]: It is a sequential recommendation method enhanced by subsequences, while also considering the intention prototype of users.
- **BSARec** [35]: It is a sequential recommendation method that incorporates an attentive inductive bias.

**(2) Proactive Recommendation Methods (ProactRec)**

- **IRN** [48]: It is a Transformer-based proactive recommendation method using a personalized impression mask generating a sequence of middle items.
- **IPG** [3]: It devises an iterative preference guidance (IPG) framework which can conduct proactive recommendation task.

## A.5 The in-depth analysis of two click simulation tactics

In this section, we provide the comparative results of the LLM-based and distribution-based click simulations to understand the power of LLM from quantitative as well as qualitative angles so that the scalability of our proposed method.

**(1) The quantitative comparison between two click modules.** We compare the distribution-based and LLM-based click simulation schemes on Lastfm and Douban_movie datasets. In this experiment settings, the nudging round $R = 10$. And evaluation metric window size $P \in [2, 4, 6, 8, 10]$. The number of target items $N_{tar}$ is 20 for Lastfm, 10 for Douban_movie dataset. The results are shown in Figure 7.

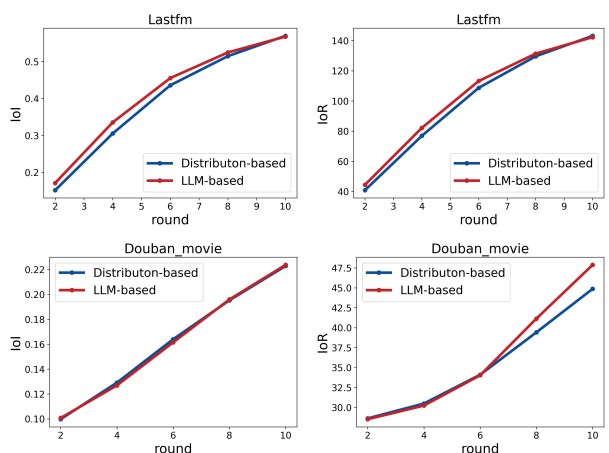

**Figure 7: The comparative results of the distribution-based and LLM-based click simulations on the Lastfm and Douban_movie datasets.**

From the figure, we can observe that the LLM-based click model yields better results than the distribution-based approach in nudging metric $IoI@P$, and $IoR@P$.

**(2) The qualitative comparison between two click modules.** To more intuitively understand the difference between the distribution-based click model and the LLM-based one, we provide a real-world case on the ML-100k dataset, which is presented in Table 5.

In the table, for the sake of compactness in displaying the sequences of intermediate items, we abbreviate the category of each item (movie) to the first three letters of its English term. For example, $Action \rightarrow Act$, $Adventure \rightarrow Adv$, $Animation \rightarrow Ani$, $Children's \rightarrow Chi$, $Comedy \rightarrow Com$, $Crime \rightarrow Cri$, $Drama \rightarrow Dra$, $Horror \rightarrow Hor$, $Mystery \rightarrow Mys$, $Romance \rightarrow Rom$, $Sci-Fi \rightarrow Sci$, $Thriller \rightarrow Thr$, $War \rightarrow War$.

Then, we select *Sci-Fi* as the target category, and use pre-match to generate 3 target items in this category. The user's latest 5 viewing movies include *Drama*, *Animation*, *Children's*, *Comedy*, and *War*. When recommending the first target movie "Robert A. Heinlein's The Puppet Masters", the LLM agent initially accepted a comedy-horror film, then tried a combination of *Action* and *Thriller* movies, as the mood of *Horror* and *Thriller* movies is generally quite terrifying. Finally, it tried the *Sci-Fi* movie "Forbidden Planet", which, although categorized only as *Sci-Fi* in the ML-100k dataset, actually also contains elements of *Action*, *Thriller*, and *Adventure*. These additional pieces of information can be captured through the powerful external information of large language models (LLM). Comparatively, the distribution-based click model first tried a few *Dramas* and *Romance* films based on the user's historical preferences, then simulated clicks on *War* movies based on the user's historical liking for this genre of films. After that, based on the similarity between *War* and *Action* movies, it clicked "Best of the Best 3", and finally accepted "Strange Days", which includes tags of *Sci-Fi* and *Action*. The guidance process for the second target movie "Aliens" was similar.

However, for the third target movie "Mars Attacks", the distribution-based method tried several rounds without landing on *Sci-Fi* movies; while the LLM-based method first followed the user's past interests by clicking on *Drama*, and *Animation* movies. Although "Balto" is tagged as an animated movie, it also includes the protagonist's adventures in the film. Therefore, the next step simulated the user to accept the *Thriller* movie "Red Rock West", which combines adventure elements. In addition to this, the movie's plot includes assassins and criminal elements. Finally, it simulates that the user will click on the intermediate movie "Dangerous Minds", which includes elements of *Action*, *Crime*, and *Sci-Fi*, and the previous leading path included some *Crime* and *Action* elements, thus the user is likely to click on the target item.

Those results affirm the effectiveness of the LLM agent in our semi-simulation environment which allows for more complex factors to be taken into account when simulating user click feedback.

## A.6 Case Study

To further illustrate the effectiveness of ITMPRec, we conduct a case study visualizing the specific user's embedding evolution to indicate the target-driven nudging process. In particular, we chose one specific user from the Lastfm dataset. The result is shown in Figure 8. From the figure, it can be seen that the user's preferences are drawn towards the target item through a round-by-round proactive recommendation process by ITMPRec.

## A.7 Interesting directions for future work

On the one hand, we will further study the causal theory [14] in the nudging process to enhance the model's explainability [24] and robustness in complex probabilistic modeling in the near future. On the other hand, we can build upon the incremental recommendation [8, 42] framework, which is now mainly used in one-round next-item recommendation tasks, to explore the trade-off between more precise results and better efficiency in proactive recommendation.

Received 14 October 2024; revised 14 October 2024; accepted 14 October 2024

**Table 5: A case study of user's click simulation to intermediate items under LLM-based and distribution-based strategies.**

| Target movies in target category *Sci-Fi* | Description |
| --- | --- |
| [1] Robert A. Heinlein's The Puppet Masters | *Sci-Fi, Horror.* |
| [2] Aliens | *Sci-Fi, Action, Thriller.* |
| [3] Mars Attacks! | *Sci-Fi, Action, Comedy, War.* |

| **The latest five movies' categories in the viewing history:** | *Drama, Animation, Children's, Comedy, War* |
| --- | --- |

| Intermediate items by LLM agent | Intermediate items by distribution-based scheme |
| --- | --- |
| Frighteners(*Com, Hor*) → Hunt for Red October(*Act, Thr*) → Forbidden Planet (*Sci*) ✓ | Breakfast at Tiffany's (*Dra, Rom*) → While You Were Sleeping (*Com, Rom*) → Great Escape (*War*) → Best of the Best 3: No Turning Back (*Act*) → Strange Days (*Sci, Act, Cri*) ✓ |
| House Party 3 (*Com*) → Dumb & Dumber (*Com*) → Star Trek IV (*Act, Adv, Sci*) ✓ | Forget Paris (*Com, Rom*) → G.I. Jane (*Act, Dra, War*) → Great Dictator (*Com*) → Star Trek IV (*Sci*) ✓ |
| Drunks (*Dra*) → Balto (*Ani,Chi*) → Red Rock West (*Thr*) → Canadian Bacon (*Com, War*) → Dangerous Minds (*Dra*) → Strange Days (*Act, Cri, Sci*) ✓ | Dangerous Ground (*Dra*)→ Hour of the Pig (*Dra, Mys*) → Red Rock West (*Thr*)→ Canadian Bacon (*Com, War*)→ Moonlight and Valentino (*Dra, Rom*) → Dangerous Minds (*Dra*) → Hunt for Red October (*Act, Thr*) |

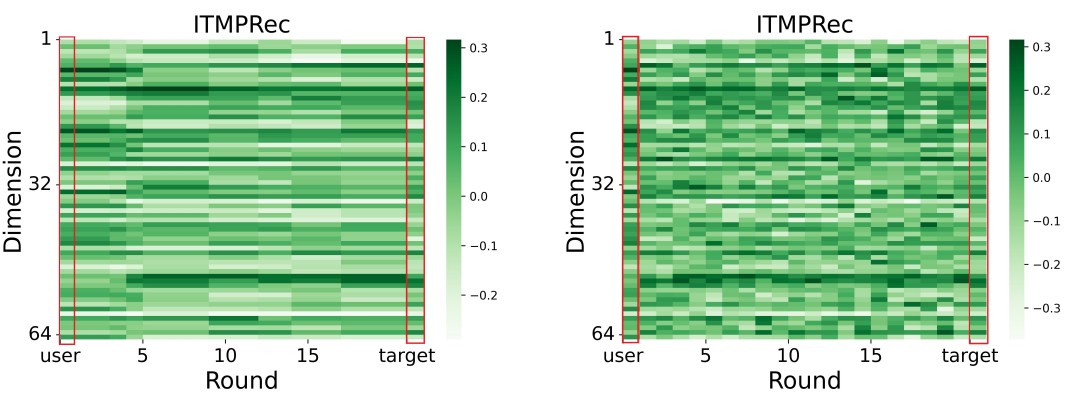

(a) user embedding's evolution in ITMPRec

(b) intermediate items recommended by ITMPRec

**Figure 8: The left sub-figure shows the embedding evolution of user** 799 **with target item** 2556**. The right sub-figure shows the embeddings of recommended items under ITMPRec. The first column of each sub-figure is the user's initial embedding and the last column shows the target item's embedding.**

