# OpenReview forum: "ITMPRec: Intention-based Targeted Multi-round Proactive Recommendation"
_ACM.org/TheWebConf/2025/Conference — WWW 2025 Poster_

### Official Review · Reviewer_LZK5 · 2024-11-11

**Novelty:** 4
**Technical Quality:** 4

**Review:**

This paper focuses on a novel proactive recommendation task in which the recommender system actively nugdes users towards specific target items. The proposed ITMPRec involves multiple modules to select target items from specific categories, as well as a multi-round simulation environment. Experimental results validate the effectiveness of ITMPRec in nudging user's preference under simulation environments.

Pros.
+ ITMPRec has achieved significant improvement on IoI and IoR metrics.
+ The paper is generally well written and easy to follow.

Cons.
- Though there have been a few works discussing proactive recommendation, the practical significance of the simulation-based model evaluation remains questionable. An LLM-based click agent is introduced to ITMPRec, however, existing works on LLM-based recommender [1] suggests that LLM could only bring limited recommendation improvement, making it doubtful as an effective simulation of real-world behavior.
- The methods are evaluated on four relatively small datasets, with some containing fewer than 1,000 users (ML-100K and Lastfm). It would be more persuasive to include other larger datasts such as ML-1M and subsets of the Amazon dataset, which is also the common practice in sequential recommendation researches.
- The backend model of ITMPRec, i.e. GraphAU is not introduced as a baseline for comparison.
- There are some typos and misnotations in the paper, such as the definition of 𝑛𝑢𝑑𝑔𝑒 score in line 462.
- The selection of $L_{N_{can}}$ is not explained in the paper.

[1] Representation Learning with Large Language Models for Recommendation. WWW 2024

**Questions:**

1. How does ITMPRec select the specific category $L_{N_{can}}$ of items?
2. From the overall comparison in table 4, it seems that IMPRec occasionally outperforms sequential recommendation baselines on HR@K metric. Since the target of ITMPRec is to distract user interests into other specific items, how could this be?

**Reviewer Confidence:**

3: The reviewer is confident but not certain that the evaluation is correct

**Scope:**

3: The work is somewhat relevant to the Web and to the track, and is of narrow interest to a sub-community

---

### Official Review · Reviewer_zD5H · 2024-11-26

**Novelty:** 3
**Technical Quality:** 3

**Review:**

The paper proposes ITMPRec, an intention-based targeted multi-round proactive recommendation method. The approach combines pre-matching, intention-induced scoring, and targeted individual arousal coefficients (TIAC) to guide user preferences progressively toward specific target items. It integrates an LLM-based click simulation agent as an alternative to traditional distribution-based click models.

Concerns and Questions:

1. While the proposed ITMPRec framework incorporates a large language model (LLM) for user feedback simulation, it does not compare its performance with existing LLM-driven recommendation models. Given the rising use of LLMs in recommendation systems, such a comparison would have provided a clearer context for ITMPRec's relative advantages or limitations.

2. The methodology relies on relatively straightforward techniques, such as the inner product for interaction tendency and linear preference updates through TIAC. While these are effective in capturing basic user behavior dynamics, they lack the complexity or sophistication often seen in state-of-the-art proactive recommendation methods, potentially limiting their ability to model nuanced user interactions.

3. The proposed model underperforms on HR@5 and HR@10 across multiple datasets compared to certain baselines. This calls into question ITMPRec's ability to deliver high-quality recommendations in early rounds, which is crucial for user satisfaction in practical applications.

**Questions:**

Refer to the Review.

**Reviewer Confidence:**

4: The reviewer is certain that the evaluation is correct and very familiar with the relevant literature

**Scope:**

3: The work is somewhat relevant to the Web and to the track, and is of narrow interest to a sub-community

---

### Official Review · Reviewer_YFKe · 2024-11-26

**Novelty:** 4
**Technical Quality:** 5

**Review:**

This paper introduces an intention-based targeted multi-round proactive recommendation method, named ITMPRec,  ITMPRec generates different intermediate items for each user during the guidance through a multi-round, progressive recommendation manner, gradually steering the user’s preferences towards the target contents.

Pros:
1.	Proactive Recommendation is a very interesting topic for recommendation.
2.	Experiments show the effectiveness of the proposed method for hit ratio in multi-round and the quality of proactive recommendations.

Cons:
1.	The paper is not easy to follow. Some claims in this paper are very abstract and make it hard to understand. For example, “by randomly designating target items within the global inventory, the chosen target might be too scattered”. What the term “scattered” means and its advantages remain unclear, and Fig (b,c) also confuses me although I carefully read the paper and figures many times.
2.	In Section 4.2 Pre-match module, it is unclear the priority of the proposed strategy. There is no theory analysis or real-world intuition for Equation (7).
3.	In Section 4.4 Targeted individual arousal coefficients, how the arousal value engages in LLM-based click agent is not clear. Only the evidence for arousal value utilization is in Appendix A.3, however, click(*,$\beta$) designed with LLM is not clear in the prompt. If it is mentioned somewhere, please highlight it in this paper.
4.	In this paper, it seems that only one item is provided as the intermediate item in the multi-round recommendation, which may be inconsistent with real-world scenarios.
5.	Simply adopting LLM as a user agent is a trivial solution for proactive recommendation.

**Questions:**

See weakness.

**Ethics Review Description:**

No Ethics Issue.

**Reviewer Confidence:**

2: The reviewer is willing to defend the evaluation, but it is likely that the reviewer did not understand parts of the paper

**Scope:**

4: The work is relevant to the Web and to the track, and is of broad interest to the community

---

### Official Review · Reviewer_GEaM · 2024-11-27

**Novelty:** 6
**Technical Quality:** 5

**Review:**

This paper proposes a novel recommendation system method, Intention-based Targeted Multi-round Proactive Recommendation (ITMPRec), aimed at addressing limitations in traditional recommendation systems, such as the entrenchment of user preferences and limited content exposure. By proactively guiding users to explore new content, the method seeks to enhance the attractiveness of specific item categories. Overall, this paper provides a new solution to the proactive recommendation problem by capturing user preferences through multiple computations of intermediate user feedback.

Pros
1.Innovative Contribution: The ITMPRec method is highly innovative in the domain of proactive recommendation. It incorporates user intentions into the recommendation process, employing mechanisms such as intention induction scores and targeted individual activation coefficients to simulate the dynamic evolution of user preferences.
2.LLM-Driven Simulation: The use of a large language model (LLM) as a click simulation agent replaces traditional distribution-based click models, providing a more sophisticated approach to capturing user feedback.
3.Comprehensive Experiments: The paper presents extensive experimental validation on multiple real-world datasets, including thorough ablation studies to illustrate the contribution of each module. Key hyperparameters are also discussed, enhancing the credibility of the results.
4.Clarity and Specificity: The ITMPRec framework is presented with a clear and detailed description, offering a well-structured and comprehensive methodology.


Cons
1. Model complexity: the method involves multiple modules and parameters, which increases computational cost and implementation complexity. Whether the user experience can be guaranteed in large-scale real-time recommendation.
2.Inconsistencies in Metrics: In proactive recommendation comparisons, the IOR metric for IRN exhibits significant discrepancies compared to other methods, whereas the HR metric performs consistently well. Given that the method's intention modeling is based on ICLRec, its moderate performance in ablation studies raises concerns.
3.Formatting Issues: The \cut\{X,num\} in Section 4.2 is not properly right-aligned, reflecting minor lapses in formatting.

**Questions:**

1. Does the proactive recommendation framework in this paper share conceptual similarities with reinforcement learning?
2. Is there a problem with using the IRN as a baseline for comparison? The results differ from those in the original paper, which appear to be too high.
3. After the model is trained, does user interest need to be recalculated before each recommendation round? If so, does this process lead to significant delays? How is low latency ensured for online recommendations?

**Reviewer Confidence:**

4: The reviewer is certain that the evaluation is correct and very familiar with the relevant literature

**Scope:**

4: The work is relevant to the Web and to the track, and is of broad interest to the community

---

### Official Review · Reviewer_kLzQ · 2024-11-30

**Novelty:** 5
**Technical Quality:** 4

**Review:**

This article proposes a novel intention-based targeted multi-round proactive recommendation method, dubbed ITMPRec. The model design is reasonable, modeling the dynamic evolution of user preferences during multiple rounds of active recommendation, and providing an alternative LLM agent to simulate real user feedback, replacing the traditional distribution based click model. The effectiveness of the model is fully demonstrated through experiments.

Weakness:
1. There are issues with unclear expression in the text, such as e_j^T∙e_u^r>e_j^T∙e_u^0. What is the reason for setting this constraint?
2. For the dataset used, there is only data statistics and a lack of description.
3. It is currently unclear how the model handles the issue of recommendation accuracy. The method of gradually directing user preferences towards the target content through multiple rounds of progressive recommendation may lead to deviation of the recommended content, which is not conducive to the accuracy of the recommendation.
4. In Figures 4 and 5, the curves on the Steam and Douban datasets are still rising. Why stop at 20 and 256 respectively?
5. How is the pre-defined value N_{tar} determined and adjusted?
6. How to explain the constraint condition of e_{idx} in formula 14?
7. Writing needs to improve, which brings trouble to the understanding.

**Questions:**

The description of the details in this article needs to be improved to some extent. Vague descriptions may lead reviewers to overlook some important contributions.

**Reviewer Confidence:**

4: The reviewer is certain that the evaluation is correct and very familiar with the relevant literature

**Scope:**

4: The work is relevant to the Web and to the track, and is of broad interest to the community